# CLAD-ST: Contrastive Learning with Adversarial Data for Robust Speech Translation

**Sathish Reddy Indurthi, Shamil Chollampatt,**
**Ravi Agrawal, Marco Turchi**
Zoom Video Communications
sathishreddy.indurthi@zoom.us, shamil.chollampatt@zoom.us,
ravi.agrawal@zoom.us, marco.turchi@zoom.us

## Abstract

The cascaded approach continues to be the most popular choice for speech translation (ST). This approach consists of an automatic speech recognition (ASR) model and a machine translation (MT) model that are used in a pipeline to translate speech in one language to text in another language. MT models are often trained on well-formed text and therefore lack robustness while translating noisy ASR outputs in the cascaded approach, degrading the overall translation quality significantly. We address this robustness problem in downstream MT models by forcing the MT encoder to bring the representations of a noisy input closer to its clean version in the semantic space. This is achieved by introducing a contrastive learning method that leverages adversarial examples in the form of ASR outputs paired with their corresponding human transcripts to optimize the network parameters. In addition, a curriculum learning strategy is then used to stabilize the training by alternating the standard MT log-likelihood loss and the contrastive losses. Our approach achieves significant gains of up to 3 BLEU scores in English-German and English-French speech translation without hurting the translation quality on clean text.

## 1 Introduction

Neural machine translation (NMT) has made significant advancements over the past several years with claims of achieving 'human parity' (Hassan et al., 2018), and 'super-human performance' (Barrault et al., 2019). However, despite making tremendous progress in quality and coverage (Costa-jussà et al., 2022), NMT models have been identified to lack robustness in dealing with noisy inputs (Belinkov and Bisk, 2018).

Robustness is especially important in *cascaded* speech translation (ST) systems, where an NMT model works on the output of the upstream automatic speech recognition (ASR) system. In this scenario, significant MT performance degradation has

| |
|---|
| **Human Transcript:** I'm not sure that's wise, given the importance of the problem, but there's now the geoengineering discussion about*: Should* that be in the back pocket in case things happen faster*, or* this innovation goes a lot slower than we *expect?* |
| **ASR Output:** I'm not sure that's why it's given the importance of the problem. But now that the geoengineering discussion about *should* that be in the back pocket in case things happen faster *or* this innovation goes a lot slower than we *expect.* |

Table 1: Example human transcript and OpenAI Whisper ASR output from the MuST-C En-De devset. The underlined text indicates erroneous phrases and *italicized* indicates punctuation errors.

been measured due to *i) error propagation* from the ASR and *ii)* the mismatch between *training-testing* condition as the NMT model is typically trained on well-formed text making it weak in dealing with noisy inputs. For these reasons, there has been significant effort towards building end-to-end ST models (Bérard et al., 2016; Weiss et al., 2017). However, due to the lack of sufficient speech translation datasets in many languages and the flexibility to independently optimize ASR and MT systems, cascaded systems continue to be a dominant approach (Anastasopoulos et al., 2022).

Prior research has tried to tackle the robustness problem in NMT models independently by (1) synthetic noise injection (Belinkov and Bisk, 2018), including mimicking ASR errors for the ST task (Sperber et al., 2017; Li et al., 2018; Martucci et al., 2021; Wang et al., 2022) or (2) by adding noise to the continuous-space representations learned by the NMT model (Sato et al., 2019; Cheng et al., 2020; Wei et al., 2022). Some of the synthetic noise injection methods replace words with probable words from ASR confusion lattices (Martucci et al., 2021; Wang et al., 2022) but fail to consider the full context and to realistically replicate ASR errors, like the phrase-level errors and segmentation errors that ASR models often make (see Table 1). On the other

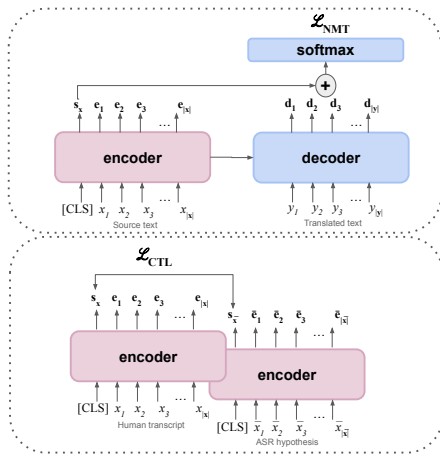

Figure 1: Proposed model and the two alternating training objectives. $\mathscr{L}_{\text{NMT}}$ is the standard NMT negative log-likelihood loss and $\mathscr{L}_{\text{CTL}}$ is the contrastive loss for improving robustness to ASR outputs.

hand, the latter approaches predominantly generate arbitrary or random error distributions and overlook the real-world error signals from the actual upstream ASR model.

To address the robustness problem, particularly in the context of cascaded ST, we propose to combine the best of the two approaches. This is obtained by training the NMT model with adversarial examples generated from the ASR outputs and encouraging the encoder representations of both the ASR outputs and their corresponding human transcripts to be closer to each other. This is done via *contrastive learning* (Saunshi et al., 2019), in addition to the standard log-likelihood NMT training, and through *curriculum* learning (Bengio et al., 2009) by pretraining on log-likelihood loss followed by iteratively alternating between the two objectives. Our method leads to significant improvements in the ST task on two language pairs (En-De, En-Fr) without hurting the performance on MT of well-formed texts. Our approach also follows the standard cascade approach that does not rely on any end-to-end ST data (speech aligned with their transcription and translation) unlike that of Bahar et al. (2021) and Lam et al. (2021), making it easier to apply to a wider range of language pairs.

## 2 Method

Our NMT model is a Transformer model (Vaswani et al., 2017) with several encoder and decoder layers, with self-attention modules within each block and a cross-attention module to link the encoder and decoder components. Training the network parameters $\Theta$ is done by minimizing the mean negative log-likelihood over a batch of $n$ source-target text translation pairs $(X, Y) = (\mathbf{x}^{(1)}, \mathbf{y}^{(1)}), \ldots, (\mathbf{x}^{(n)}, \mathbf{y}^{(n)})$. This objective encourages producing a target sentence $\mathbf{y}$ with a sequence of tokens $y_1, y_2, \ldots, y_{|\mathbf{y}|}$ given the source sentence $\mathbf{x}$ with sequence of tokens $x_1, x_2, \ldots, x_{|\mathbf{x}|}$.

$$\mathscr{L}_{\text{NMT}} = -\sum_{i=1}^{n} \sum_{j=1}^{|\mathbf{y}^{(i)}|} \log P(y_j^{(i)} | \mathbf{x}^{(i)}, \mathbf{y}_{<j}^{(i)}, \Theta)$$

To improve the robustness of the NMT model on noisy ASR outputs for cascaded speech translation, we use a *contrastive* learning method (Wei et al., 2021) that is aimed to bring the representations of a noisy sentence generated by the upstream ASR closer to its clean version (the human transcript for the same audio) in the semantic space modeled by the NMT encoder.

To get the encoder sentence representations efficiently for contrastive learning, a [CLS] token is prepended to the input sentences similar to the BERT model (Devlin et al., 2019). We found that using encoder output corresponding to the [CLS] token is slightly better than using the mean of encoder outputs as the sentence representation. $\mathbf{s_x}$, which also gets added to each of the decoder outputs before softmax (see Figure 1).

Contrastive learning uses speech transcription corpora (i.e., speech paired with human transcripts). The speech input is passed through the upstream ASR model to obtain the ASR outputs. Given the noisy ASR output $\bar{\mathbf{x}}$ paired with its corresponding clean human transcript $\mathbf{x}$, we minimize the contrastive objective $\mathscr{L}_{\text{CTL}}$, which is an average of two symmetric sentence-level contrastive loss functions.

Given a batch of $n$ examples $(X, \bar{X}) = \{(\mathbf{x}^{(1)}, \bar{\mathbf{x}}^{(1)}), \ldots, (\mathbf{x}^{(n)}, \bar{\mathbf{x}}^{(n)})\}$

$$\mathscr{L}_{\text{CTL}} = \frac{1}{2n} \sum_{i=1}^{n} \left( \mathcal{L}_c(\mathbf{x}^{(i)}) + \mathcal{L}_c(\bar{\mathbf{x}}^{(i)}) \right) \quad (1)$$

where

$$\mathcal{L}_c(\mathbf{x}) = \frac{-\exp\left(D(\mathbf{s_x}, \mathbf{s_{\bar{x}}})\right)}{\exp\left(D(\mathbf{s_x}, \mathbf{s_{\bar{x}}})\right) + \sum_{x'} \exp(D(\mathbf{s_x}, \hat{\mathbf{s}}_{x'}))}$$

$D(\mathbf{u}, \mathbf{v})$ denotes the cosine distance between two vectors $\mathbf{u}$ and $\mathbf{v}$. $\hat{\mathbf{s}}_{x'}$ represents a negative example

constructed for every other sentence $\mathbf{x}'$ in the batch. Given $\mathbf{x}' \in (X \cup \bar{X}) \setminus \{\mathbf{x}, \bar{\mathbf{x}}\}$ and its sentence embedding $\mathbf{s}_{x'}$

$$\hat{\mathbf{s}}_{\mathbf{x}'} = \begin{cases} \mathbf{s}_{\mathbf{x}} + \lambda_{\mathbf{x}}(\mathbf{s}_{\mathbf{x}'} - \mathbf{s}_{\mathbf{x}}) & \text{if, } d_{\mathbf{x}}^+ > d_{\mathbf{x}}^- \\ \mathbf{s}_{\mathbf{x}'} & \text{otherwise} \end{cases} \quad (2)$$

where $d_{\mathbf{x}}^+ = \|\mathbf{s}_{\mathbf{x}} - \mathbf{s}_{\bar{\mathbf{x}}}\|_2$, $d_{\mathbf{x}}^- = \|\mathbf{s}_{\mathbf{x}} - \mathbf{s}_{\mathbf{x}'}\|_2$, and $\lambda_{\mathbf{x}} = \frac{d_{\mathbf{x}}^+}{d_{\mathbf{x}}^-}$. The above linear interpolation in Eq 2 with exponentially decaying $\lambda_{\mathbf{x}}$ is implemented following Wei et al. (2021) to increase the hardness of negative examples by not selecting uninformative negative examples as training progresses. $\mathcal{L}_c(\bar{\mathbf{x}})$ follows similar formulation as $\mathcal{L}_c(\mathbf{x})$ with $\mathbf{x}$ replaced by $\bar{\mathbf{x}}$, and vice versa.

To stabilize the NMT parameters before applying contrastive learning, we also use curriculum learning (Bengio et al., 2009). We first pre-train the model with NMT loss $\mathcal{L}_{\text{NMT}}$ only for the first $N$ batches, followed by iteratively training alternate batches with $\mathcal{L}_{\text{CTL}}$ and $\mathcal{L}_{\text{NMT}}$ losses, respectively, until convergence. With this model and training regime, which alternates between training with translation and speech recognition datasets independently, we are able to build robust NMT models for cascaded ST without any requirement for speech translation data.

## 3 Experimental Setup

We experiment with English-German (En-De) and English-French (En-Fr) language directions.
**Training Data**: For parallel text translation data, we use WMT'16 En-De (Bojar et al., 2016) and WMT'14 En-Fr (Bojar et al., 2014) news translation task data for the corresponding language directions. To get the ASR data for contrastive learning, we use Open AI Whisper base ASR (Radford et al., 2022) to transcribe the speech in the training sets from Mozilla Common Voice 12.0 (Ardila et al., 2020), VoxPopuli (Wang et al., 2021), and the English set of Multilingual LibrSpeech (Pratap et al., 2020) for both language directions. As an additional experiment, we also try adding in-domain MuST-C training data. The full statistics of the datasets are in Appendix A.2 (Table 8).
**Evaluation**: We use MuST-C (Di Gangi et al., 2019) tst-COMMON as our test set and MuST-C dev as our validation set. For MT and ST tasks, the inputs to the model are the Whisper (base) ASR outputs and the corresponding human transcripts,

| NMT | En-De | | En-Fr | |
|---|---|---|---|---|
| | MT | ST | MT | ST |
| Transformer-base | 28.1 | 23.5 | 36.1 | 30.8 |
| CONF-ASR (Wang et al., 2022) | 28.2 | 25.2 | 36.3 | 32.0 |
| CSA-NMT (Wei et al., 2021) | **29.5** | 24.4 | **37.2** | 31.3 |
| CLAD-ST | 28.8 | **26.6** | 36.5 | **33.4** |

Table 2: Performance of various models on MT and ST tasks on MuST-C tst-COMMON.

respectively. We evaluate against the English and French reference translations for the respective translation directions. We use case-sensitive, detokenized BLEU computed using sacrebleu[1] (Post, 2018). We also compute statistical significance using paired bootstrap resampling (Koehn, 2004).
**Models**: We use a base Transformer (Vaswani et al., 2017) as our baseline NMT model, which we extend to support our proposed approach, Contrastive Learning with Adversarial Data for Speech Translation (**CLAD-ST**). We also compare against two recent state-of-the-art approaches for NMT robustness which we re-implement: (1) **CONF-ASR** (Wang et al., 2022) generates adversarial examples by substituting random source tokens with candidate tokens from ASR confusion set based on their embedding distances to the original token. (2) **CSA-NMT** (Wei et al., 2022)[2] uses a separate semantic encoder to model a semantic region around the source and target sentences. To improve NMT robustness, samples from this region are used to augment the NMT decoder representations.

We tokenize all datasets using a sentencepiece model (Kudo and Richardson, 2018) for each language pair on the text translation datasets with a shared source-target vocabulary size of 32,000. We use OpenAI Wispher (base) ASR as our main upstream model. We also evaluate the ST performance of our model using Whisper large ASR. Other training details and hyperparameters are in Appendix A.1.

## 4 Results and Analysis

### 4.1 Overall Performance

Our proposed CLAD-ST achieves 26.6 and 33.4 BLEU on the En-De and En-Fr ST tasks, significantly ($p < 0.001$) outperforming the Transformer baseline by 3.1 BLEU points and 2.6 BLEU

---

[1]https://github.com/mjpost/sacrebleu. Signature: nrefs:1|case:mixed|eff:no|tok:13a|smooth:exp
[2]Code and network hyperparameters are from https://github.com/pemywei/csanmt

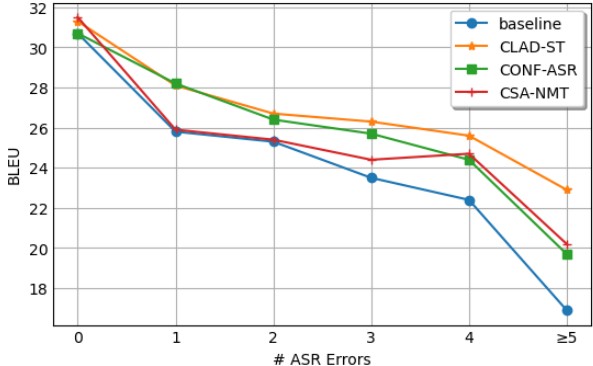

Figure 2: BLEU scores compared to the number of token-level errors in the ASR outputs on En-De ST task on MuST-C tst-COMMON dataset.

points (see Table 2). Our approach consistently performs better than other baselines (CONF-ASR and CSA-NMT), yielding gains of 1.4 BLEU points ($p < 0.001$) on both language directions. The improvement of our approach over CONF-ASR shows the importance of contextual modeling of the ASR errors instead of simulating them with ASR confusion lattices. CSA-NMT achieves the largest improvement in the MT task but fails to specialize to ASR errors, yielding lower improvements compared to our approach.

## 4.2 Robustness

In Figure 2, we report the performance of all the En-De models as the number of ASR errors increases in the input sequence. We group the ASR outputs based on the number of tokens different from the reference transcripts on the test set. The performance of the Transformer baseline model drastically drops as the number of ASR errors increases. CONF-ASR performs better than the Transformer baseline in all the ASR error conditions and closely with CLAD-ST on ASR outputs with one or two errors. However, CLAD-ST notably outperforms it as the number of errors increases. CSA-NMT performs slightly better than the baseline Transformer when the number of errors is few and performs slightly better than CONF-ASR when increasing the number of errors due to their semantic augmentation. Overall, our approach has shown to be robust against sequences having several ASR errors and performs consistently better than the other baselines, irrespective of the number of errors in the sequence.

## 4.3 Curriculum Learning

We study the effect of the curriculum learning strategy (Section 2) and reports the results in Table 3. We find that curriculum learning strategy brings an improvement of 0.5 BLEU on En-De and 0.7 BLEU On En-Fr ST tasks, respectively.

| Model | En-De | En-Fr |
|---|---|---|
| Transformer-base | 23.5 | 30.8 |
| CLAD-ST | 26.6 | 33.4 |
| w/o Curriculum learning | 26.1 | 32.7 |

Table 3: Performance on ST tasks with and without curriculum learning on MuST-C tst-COMMON.

## 4.4 Out-of-distribution ASR

In Table 4, we report the performance of all the En-De systems using Whisper base as the ASR model during training and a higher-quality ASR (Whisper large) during testing. Although training on lower-quality ASR outputs with more errors, a similar BLEU score improvement is also noticed for all the systems when fed better ASR outputs. However, CLAD-ST achieves the largest improvements over the baseline, showing that it better generalizes over different ASR-quality outputs.

| Model | En-De | |
|---|---|---|
| | Wbase (WER = 14.3) | Wlarge (WER = 7.6) |
| Transformer-base | 23.5 | 24.4 |
| CONF-ASR (Wang et al., 2022) | 25.2 | 26.3 |
| CSA-NMT (Wei et al., 2022) | 24.4 | 25.8 |
| CLAD-ST | **26.6** | **27.4** |

Table 4: BLEU scores on MuST-C En-De tst-COMMON ST task when using Whisper-large (Wlarge) vs. Whisper-base (Wbase) ASR. Word error rate (WER) of the ASR model is provided.

## 4.5 Model Size

To investigate if our results scale when we increase the size of the network, we use Transformer-large instead of Transformer-base for our baseline models as well as CLAD-ST. The results are reported in Table 5. We find that we get improvements of 1.5 and 1.3 BLEU score points for the ST task on En-De and En-Fr, respectively, compared to CONF-ASR, demonstrating its advantage even on larger NMT architectures.

## 4.6 In-domain Training Data

We also experiment with adding in-domain training data from MuST-C for training the NMT models in

| NMT | En-De | | En-Fr | |
|---|---|---|---|---|
| | MT | ST | MT | ST |
| Transformer-big | 29.6 | 24.9 | 38.7 | 33.3 |
| CONF-ASR (Wang et al., 2022) | 29.8 | 26.6 | 38.8 | 34.6 |
| CSA-NMT (Wei et al., 2021) | **31.1** | 25.7 | **39.9** | 33.7 |
| CLAD-ST | 30.3 | **28.1** | 39.1 | **35.9** |

Table 5: Performance of using Transformer-large as the model architecture on MT and ST tasks on MuST-C tst-COMMON.

CLAD-ST and the baselines (see Table 6). In line with our expectations, we find a notable improvement in performance of all the systems. Despite the improved baselines, we observe similar improvements for CLAD-ST of 1.4 BLEU on both En-De and En-Fr ST tasks compared to CONF-ASR, and 2.4 and 2.2 BLEU on En-De and En-Fr ST tasks, respectively, compared to CSA-NMT.

| NMT | En-De | | En-Fr | |
|---|---|---|---|---|
| | MT | ST | MT | ST |
| Transformer-base | 31.0 | 26.1 | 37.9 | 32.9 |
| CONF-ASR (Wang et al., 2022) | 30.9 | 27.9 | 38.0 | 33.9 |
| CSA-NMT (Wei et al., 2021) | **32.1** | 26.9 | **38.8** | 33.1 |
| CLAD-ST | 31.6 | **29.4** | 38.2 | **35.3** |

Table 6: Performance of various models on MT and ST tasks on MuST-C tst-COMMON when adding in-domain MuST-C training data.

## 5 Related Work

Several prior works tried to improve the robustness of MT models to noisy ASR inputs for cascaded speech translation. Sperber et al. (2017) randomly sampled synthetic noise in the form of insertions, deletions, and substitutions and injected them into the source side of MT training data. Li et al. (2018) encoded the syllables (*Pinyin*) for Chinese ST and injected synthetic character-level substitutions to simulate ASR errors. In contrast, Martucci et al. (2021) trained a *lexical noise model* using confusion sets extracted by aligning the actual ASR output to reference transcript to inject synthetic errors. Wang et al. (2022) picked substitution candidates from the ASR confusion set that are farthest away from the source token embeddings leading to more robust MT models. Cheng et al. (2019) also used the actual ASR output in training incorporating an additional discriminator network that tries to distinguish between human transcript and the ASR output following Cheng et al. (2018) to improve robustness. Padfield and Cherry (2021) simulated not only token-level errors but also segmentation errors

and projected them on the target side as well. Bahar et al. (2021), on the other hand, proposes to jointly optimize the ASR and MT components in the cascaded system using speech-translation data. Similarly, Lam et al. (2021) also uses speech-translation data to reduce error propagation between the two components by mutually training the MT system on ASR n-best outputs and using cyclic feedback from MT outputs for self-training of the ASR model.

## 6 Conclusion

We improve the robustness of MT to ASR outputs using contrastive learning to bring the representations of clean and noisy examples closer in the semantic space. Our approach does not require any speech translation corpus. We significantly improve the translation accuracy on noisy ASR outputs without degrading translation accuracy on clean text. We also show that the approach is scalable to better-quality ASR models in the cascade other than the one used during training. The proposed approach is generic and is applicable beyond the context of speech translation alone, such as translating user-generated chat text or non-native text if paired noisy-clean data is available.

## Limitations

The limitations of our paper are:

- As in any ST cascade architecture, the performance of our MT system depends on the quality of the ASR model outputs. Since we use examples generated by the ASR to train the MT model, the quality of ASR outputs directly affects the model performance. In our work we tested two ASR systems having different quality (base: 14.3 WER and large: 7.6 WER). This is particularly relevant when English is not the source language side.

- Evaluation is done using only the BLEU score. We did not use human evaluation or COMET.

- The evaluation is limited to two language pairs having English as the source. This selection of the source language is quite important because having a no-English language as a source will expose our MT model to a) ASR models of probably lower quality and b) different and more varied linguistic challenges that might affect the work of the adversarial method.

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

## A Implementation Details

### A.1 Training Details and Hyperparamters

We implement our approach and (Wang et al., 2022) using Fairseq[3]. For Wang et al. (2021), we randomly selected 14% tokens of sentences for a replacement to generate adversarial examples based on the ASR word error rate. All experiments are performed on A100 GPU, and gradient accumulation and batch size are set to 8 iterations and 8k tokens. For CLAD-ST, our NMT model is pretrained for the first 50K batches for NMT loss in the curriculum. For evaluation, the beam size and length penalty are set to 4 and 0.6 for both tasks. Hyperparameters for all the models are shown in Table 7.

---

[3]https://github.com/facebookresearch/fairseq

| Hyperparameter | Baseline | CONF-ASR | CSA-NMT | CLAD-ST |
|---|---|---|---|---|
| encoder layers | | | 6 | |
| encoder embed dim | | | 256 | |
| encoder ffn embed dim | | | 2048 | |
| encoder attention heads | | | 8 | |
| decoder layers | | | 6 | |
| decoder embed dim | | | 256 | |
| decoder ffn embed dim | | | 2048 | |
| decoder attention heads | | | 8 | |
| dropout | | | 0.3 | |
| optimizer | | | adam | |
| adam-$\beta$ | | | (0.9, 0.98) | |
| clip-norm | | | 0.0 | |
| lr scheduler | | | invers-sqrt | |
| learning rate | | | 5e-4 | |
| warmup-updates | | | 4000 | |
| label-smoothing | | | 0.1 | |
| max tokens | 12800 | 8192 | 6144 | 8192 |
| num-initial-updates-nmt | N/A | 50000 | 50000 | 50000 |
| csanmt-semantic-samples | N/A | N/A | 20 | N/A |

Table 7: Model hyperparameters

| Type | Name | Lang. | #Examples |
|---|---|---|---|
| MT Data | WMT'16 | En-De | 4.5M |
| | WMT'14 | En-Fr | 40 M |
| | MuST-C | En-De | 251K |
| | train | En-Fr | 275K |
| ASR Data | VoxPopuli | En | 0.18M |
| | Commonvoice | En | 0.99M |
| | MLS | En | 11M |
| Dev | MuST-C Dev | En-De | 1415 |
| | | En-Fr | 1412 |
| Test | MuST-C | En-De | 2580 |
| | tst-COMMON | En-Fr | 2632 |

Table 8: Statistics of datasets used in the experiments.

## A.2 Datasets

Dataset statistics can be found in Table 8. WMT'16 En-De training data[4] includes Europarl, Common Crawl corpus, and News Commentary corpora. WMT'14 En-Fr[5] training data includes Europarl, Common Crawl, UN corpus, News Commentary, and Fr-En Gigaword corpus.

---

[4]https://www.statmt.org/wmt16/translation-task.html

[5]https://www.statmt.org/wmt14/translation-task.html