# OpenReview forum: "CLAD-ST: Contrastive Learning with Adversarial Data for Robust Speech Translation"
_EMNLP/2023/Conference — EMNLP 2023 Main_

### Official Review · Reviewer_1msj · 2023-07-29

**Typos Grammar Style And Presentation Improvements:** 1. A missing full stop in the caption…
**Soundness:** 3

**Excitement:**

4: Strong: This paper deepens the understanding of some phenomenon or lowers the barriers to an existing research direction.

**Paper Topic And Main Contributions:**

This paper aims at improving the robustness of NMT from noisy ASR transcriptions. The authors design a simple yet effective method based on contrastive learning to bridge the discrepancy between the sentence representation of clean texts and real transcriptions. Experimental results prove the effectiveness of the proposed method.

**Reasons To Accept:**

- The paper is well-written and easy to follow.
- The proposed method is pretty intuitive and proved to be effective on both clean texts and noisy texts.
- Further analyses prove that the method is applicable to various levels of noise.

**Reasons To Reject:**

- One of the limitations of the method is the need for human transcripts. For the languages or scenarios where human transcripts are difficult to obtain, the application of the method may be limited.
- As a contrastive learning method, it is uncertain whether the performance of this method will be affected by the noisy level in ASR transcripts during training. Will the performance of the method degrade when the noise in the training samples is enhanced? What is the best magnitude of the ASR noise in training data?


**Reproducibility:**

4: Could mostly reproduce the results, but there may be some variation because of sample variance or minor variations in their interpretation of the protocol or method.

**Reviewer Confidence:**

4: Quite sure. I tried to check the important points carefully. It's unlikely, though conceivable, that I missed something that should affect my ratings.

---

> ### Author Rebuttal · Authors · 2023-08-29
>
> Dear Reviewer 1msj,
>
> We thank you for your valuable feedback on our paper. Please find our responses below.
>
> > **_“One of the limitations of the method is the need for human transcripts.”_**
>
> Datasets containing human transcripts are the basement to build the ASR systems and they are available in large quantities and cover a large set of languages (e.g. Common Voice, Multilingual LibriSpeech). These datasets can be used without effort.  In the case of low-resource languages with limited human transcripts, their creation is possible by leveraging the work of monolingual speakers in that particular language. This makes this process much simpler than the generation of speech-translation data that, instead, requires bilingual speakers. It is important to notice that when no human transcripts are available, the main problem becomes the creation of the ASR, making the application of our method a secondary problem.
>
> > **_“Will the performance of the method degrade when the noise in the training samples is enhanced? What is the best magnitude of the ASR noise in training data?“_**
>
> To show how the noise level impacts the performance, we conducted two experiments using Whisper-base and Whisper-large models (Table 3). The Whisper-base has a WER 14.3 on our test set and Whisper-large has a WER 7.6. Training datasets constructed from these models differ by noise level. In both cases, the proposed approach outperforms the other methods by a  huge margin, however, the model training with less noisy transcripts (generated with a large model) achieves the highest performance.

---

### Official Review · Reviewer_ZhJE · 2023-08-01

**Soundness:** 4

**Excitement:**

3: Ambivalent: It has merits (e.g., it reports state-of-the-art results, the idea is nice), but there are key weaknesses (e.g., it describes incremental work), and it can significantly benefit from another round of revision. However, I won't object to accepting it if my co-reviewers champion it.

**Paper Topic And Main Contributions:**

This paper presents a method for improving cascaded speech translation, which usually lacks in robustness since the MT model has to translate error-prone ASR outputs. The authors propose use contrastive learning to train the MT model to bring the representations of the ASR model and the ground truth source text closer in the semantic space. In their experiments using Whsiper as the ASR model, they show that their method can surpass the baseline cascaded-ST and two more similar methods from the literature in two language pairs of MuST-C. They also show that the proposed method is more robust than the others as the number of errors in the ASR output increases.

**Questions For The Authors:**

A. Did you try a different approach in getting the sentence representation? Like mean or max pooling?

B. What is the impact of curriculum learning in the proposed method?

**Reasons To Accept:**

The paper is very well written and easy to follow. The proposed method is intuitive, seems to be easily integratable to the MT model training, and yields significant improvements over the baseline and the other two tested methods (CONF-ASR, CSA-NMT) in the two tested language pairs. Furthermore is more robust to more error-prone ASR transcriptions.

**Reasons To Reject:**

The experimental setup is rather limited, which is partly acknowledged in the limitations. Only two language pairs are being tested (En -> De/Fr), both of them having English source speech, and being high-resource. Additionally, the experiments are done using Whisper, which in general is a very good quality ASR system. An expanded experimental setup, using lower-quality ASR systems, pairs with non-English source, would shed more light into the usefulness of the proposed method in other non-standard scenarios.

I find the claim of the introduction (L089-L094) a bit misleading. Using a cascade ST system eliminates the need for parallel ST data, and not the proposed method itself (as claimed). Furthermore, in the second part of the same claim, it is stated that "making it easier to apply to a wider range of language pairs compared to the end-to-end approaches", but the experiments are only done in the standard pairs that E2E systems are tested, which is a missed opportunity.

**Reproducibility:**

4: Could mostly reproduce the results, but there may be some variation because of sample variance or minor variations in their interpretation of the protocol or method.

**Reviewer Confidence:**

4: Quite sure. I tried to check the important points carefully. It's unlikely, though conceivable, that I missed something that should affect my ratings.

**Typos Grammar Style And Presentation Improvements:**

L063-L068: which are these methods?

L181-L182: Is this a typo? It seems that you use tst-COMMON both for validation and testing.

L193-L194: This is the first time that the acronym is presented. Consider using the full name here.

L219: EN -> En (for consistency with the rest of the paper)

Table 2: Consider adding the statistical significance in the table and caption.

Figure 2 & L237: number of token vs number of words

Figure 2: add the tested language pair and the test set.

Table 3: This table and its caption are very confusing. Consider reformulating it. Also it is not stated what the metrics are and the tested language pair (or set).

Table 4: Caption says "statistics of datasets used for training" but also dev and test set are present.

---

> ### Author Rebuttal · Authors · 2023-08-29
>
> Dear Reviewer ZhJE,
>
> We thank you for your valuable feedback on our paper. Please find our responses below.
>
> >  _**“The experimental setup is rather limited, which is partly acknowledged in the limitations”**_
>
> We considered two English-source high-resource language pairs as they tend to produce better cascaded baselines (due to a good quality English ASR and strong NMT models) and should suffer less from error propagation compared to low-resource pairs. This makes them stronger baselines to demonstrate the effectiveness of our approach that aims to reduce this error propagation. With a non-English source low resource pair, we expect to observe similar or better improvements as they tend to suffer more from error propagation.
>
> Several prior published work on speech translation rely on these language pairs alone in their experimental setup (ex:  [Tran et al, EMNLP 2022](https://aclanthology.org/2022.emnlp-main.297.pdf) relies only on En-De MuST-C ST benchmark, [Ao et al, ACL 2022](https://aclanthology.org/2022.acl-long.393.pdf) report only on En-De and En-Fr ST tasks from the MuST-C benchmark similar to ours).
>
> >  _**“Using a cascade ST system eliminates the need for parallel ST data, and not the proposed method itself (as claimed)”**_
>
> Our work is on reducing the error propagation between the ASR and MT models and improving robustness of cascaded speech translation. Some prior work, including the two additional references pointed out by Reviewer 1, rely on parallel speech translation data to improve cascaded systems. Our approach does this without it. We will clarify the context of the claim in the revised version.
>
> **Answers to Questions**
>
> > **_A. “Did you try a different approach in getting the sentence representation?”_**
>
> We tried the mean as the sentence representation. However, [CLS] performed better than the mean (~0.1-0.2 BLEU), so we decided to select [CLS] representation for our experiments. We will include the result in the revised paper. Note that using either  [CLS] or mean representation in our approach performed better than our MT baselines.
>
> > **_B. “What is the impact of curriculum learning in the proposed method?”_**
>
> The use of curriculum learning improved the performance of our model in both language directions (+0.51 BLEU for En-De and +0.73 BLEU for En-Fr). We will add this in the revised version.
>
> **Typos Grammar Style And Presentation Improvements**
>
> We will improve the paper by incorporating the suggestions and fixing the typos. Please note that we indeed used the “dev” portion of MuST-C for validation (thanks for pointing out the typo).

---

### Official Review · Reviewer_FsQ5 · 2023-08-02

**Soundness:** 4

**Excitement:**

4: Strong: This paper deepens the understanding of some phenomenon or lowers the barriers to an existing research direction.

**Missing References:**

Since there are much recent work on cascade AST, it would be great if there is a comparison between these two methods:
1) Bahar et. al 2021: Tight integrated end-to-end training for cascaded speech translation
2) Lam et. al 2021: Cascaded Models With Cyclic Feedback For Direct Speech Translation

**Paper Topic And Main Contributions:**

This paper presents a method, called CLAD-ST, which enhances cascade speech-to-text (AST) translation via improving the robustness of its neural machine translation (NMT) to its speech recognition (ASR) outputs.

There are two training objectives for the NMT: 1) contrastive learning and 2) cross-entropy training. In contrastive learning, it takes the ASR output and the corresponding gold-reference transcriptions as inputs. Such loss aims to reduce the distance between these two sentences while pushing them away from the other sentences, or called negative examples, resulting in stronger robustness of the NMT encoder towards the ASR output. Each sentence is approximated by the representations of a "[CLS]" token that is prepended at the beginning of the source sentence. The cross-entropy training is on the NMT's parallel corpora. In order to stabilise the training, the NMT is first pre-trained on the cross-entropy loss, followed by alternative training between the two losses.

Experiments on En-De and En-Fr language pairs demonstrate the effectiveness of CLAD-ST over their baselines. It is worth noting that while the speech translation corpus "MuST-C" is used in evaluation, it is not used in training the NMT model.

Overall, CLAD-ST is a simple but effective method to improve cascade AST. It is rather data-efficient as it does not require the triplet of speech-transcription-translation corpus nor paired speech-translation data. Furthermore, the training part focuses only on the NMT with minimal intervention to the architecture of the existing NMT model. However, a possible weakness of the work would be the relatively weak (compared to the Whisper-large) pre-trained NMT model, which is a shallow transformer model.





**Questions For The Authors:**

- What is the performance of the cascade AST when MuST-C is used in training the ASR and the NMT model?

**Reasons To Accept:**

- Compared to research on direct model, there are less work about cascade AST. This paper presents a focused contribution to improve it.
- The presented method is simple but effective. It does not require triplet of speech-transcription-translation nor speech-translation paired data. It does not require sophisticated changes to the NMT architecture, improving its practicality in production environment.

**Reasons To Reject:**

- The model size of the NMT is relatively small, i.e., a transformer-based architecture. This might raise concern on the improvement brought by the contrastive loss when scaling up the NMT training.


**Reproducibility:**

4: Could mostly reproduce the results, but there may be some variation because of sample variance or minor variations in their interpretation of the protocol or method.

**Reviewer Confidence:**

4: Quite sure. I tried to check the important points carefully. It's unlikely, though conceivable, that I missed something that should affect my ratings.

**Typos Grammar Style And Presentation Improvements:**

- line 051: inflexibility <- flexibility
- The caption of Figure 1 can be more informative.
- Please also add the "signature" when reporting evaluation metrics that are implemented in sacrebleu.

---

> ### Author Rebuttal · Authors · 2023-08-29
>
> Dear Reviewer FsQ5,
>
> We thank you for your valuable feedback on our paper. Please find our responses below.
>
>  > _**”The model size of the NMT is relatively small, i.e., a transformer-based architecture. This might raise concern on the improvement brought by the contrastive loss when scaling up the NMT training.”**_
>
> We followed comparable work in cascaded speech translation (e.g. Wang et al, 2021) in choosing Transformer (base) as the baseline model. Nonetheless, we believe our approach will show improvement on larger Transformer baselines also since larger models tend to overfit more on the well-formed MT training data. Our proposed method reduces such overfitting via the additional loss term aimed to improve robustness on noisy ASR output. We will include a comparison in the revised paper using bigger Transformer models.
>
>  >  _**“What is the performance of the cascade AST when MuST-C is used in training the ASR and the NMT model?”**_
>
> Our approach does not rely on speech translation corpus and hence using a significantly smaller resource such as MuST-C to train ASR and NMT independently is expected to yield a poorer baseline cascaded system. However, we will include additional experiments in the revised version using MuST-C.
>
>  **Missing references**: We will add references to Bahar et. al 2021 and Lam et. al 2021. The key difference is that both approaches rely on speech translation data.
>
>  We will fix the typos and incorporate both suggestions for improving the presentation.

---

### Meta-Review · Area_Chair_kLDr · 2023-09-11

**Recommendation:** 5

**Metareview:**

Reviewers agree that the proposed method is simple and effective for improving NMT robustness in a cascade De|Fr-En speech-to-text translation scenario. The main concern that is not addressed by additional experiments promised in the rebuttals is the limited focus on only 2 high resource language pairs. While adding low resource language pairs would indeed strengthen the paper since it is likely to highlight the strengths of the proposed method, the reviewer agrees that this is beyond the scope of a short paper. However, the long paper format might have been a better fit.

---

### Decision · Program_Chairs · 2023-10-07

**Decision:**

Accept-Main

**Comment:**

Reviewers agree that the proposed method is simple and effective for improving NMT robustness in a cascade De|Fr-En speech-to-text translation scenario. The main concern that is not addressed by additional experiments promised in the rebuttals is the limited focus on only 2 high resource language pairs. While adding low resource language pairs would indeed strengthen the paper since it is likely to highlight the strengths of the proposed method, the reviewer agrees that this is beyond the scope of a short paper. However, the long paper format might have been a better fit.